# The Impact of Organic and Intensive Agricultural Activity on Groundwater and Surface Water Quality

**Laima Česonienė** [1,*] **, Daiva Šileikienė** [1]**, Laura Čiteikė** [1]**, Gintautas Mozgeris** [2] **and Koike Takayoshi** [3]

1  Department of Environment and Ecology, Faculty of Forest Science and Ecology, Vytautas Magnus University, Agriculture Academy, Studentų Str. 11, LT–53361 Akademija, Kaunas, Lithuania
2  Department of Forest Sciences, Faculty of Forest Science and Ecology, Vytautas Magnus University, Agriculture Academy, Studentų Str. 11, LT–53361 Akademija, Kaunas, Lithuania
3  Research Faculty of Agriculture, Hokkaido University, Sapporo 060-8589, Japan
*  Correspondence: laima.cesoniene1@vdu.lt; Tel.: +370-37-752224

**Abstract:** The poor condition of surface water is still a problem in many countries, including Lithuania. To assess the impact of organic agricultural production on groundwater and surface water quality in Lithuania, surface water samples from rivers and other bodies of water are usually studied, leaving the properties of groundwater in agricultural fields unknown. Samples of river water and groundwater collected from both organic and intensive farming fields bordering the studied rivers were investigated in this study. The study was conducted on five rivers located in the Nemunas River Basin District and in 23 cultivated neighboring fields, where wells were drilled 4–5 m deep for groundwater sampling. All five rivers corresponded to the values of good and very good in terms of their ecological status, according to the values of $PO_4$-P and $NH_4$-N. According to the total P value, one river did not correspond to the values of good or very good ecological status. According to the total N value, four rivers did not meet the values of good or very good ecological status. We found that, with the exception of one farm, the pH, total P, total N, and $NO_3$-N, as well as the concentrations of $NH_4$-N and $PO_4$-P, in the groundwater from organic farms were lower compared to the groundwater from intensive farming areas. This suggests the importance of ground water sampling in addition to surface water surveys in water quality studies related to agricultural production.

**Keywords:** pollution; ecological status indicators; water quality; groundwater; surface water

## 1. Introduction

Agricultural practices are regarded as a major factor in poor water quality across many EU member states [1], and agricultural systems and activities can burden freshwater resources [2]. A scientific evaluation of the suitability and cost-effectiveness of the options for reducing nutrient loss as a result of agricultural land near surface waters on the catchment scale, including the feasibility of the options under different climatic and geological conditions in different E.U. countries, is presented in [3]. The increasing rate of water resource use has resulted in contamination by wastewater from domestic, industrial, and agricultural sectors. Researchers have reached a consensus: agricultural pollution poses a considerable challenge to crop security and human health, especially in economically developed areas [4]. Various research works have shown that the excessive use of chemical fertilizers and pesticides causes serious non-point source water pollution [5]. Diffuse and other water pollution is a major problem in many agroecosystems, especially in irrigated areas linked to high-value ecosystems [6]. The quality of water is determined for surface and groundwater by measuring their N and P content; for the surrounding ecological areas, the N deposition is measured [7]. With global population growth and the concomitant increase in food demand, the use of fertilizers for agriculture has increased over the last few decades [8]. Despite their benefits for farm productivity and profitability, fertilizers can lead to excessive nutrient enrichment in aquatic ecosystems (e.g., rivers, streams, and

estuaries) via agricultural runoff and sewage discharge, known as eutrophication [9,10]. Fertilizers and manure applied to cropland to increase yields are often subject to surface erosion, soil leaching, and runoff, increasing nutrient loads in surface and sub-surface waters and degrading water quality [11]. Mineral exploitation, chemical enterprise operations, pesticide and fertilizer application, sewage discharge, and vehicle emissions are the pollution sources in agricultural land [4].

The use of inorganic fertilizers in large quantities to meet the N demand of crops, in combination with the return flow associated with irrigation, has increased $NO_3$ concentration in the surface water bodies and groundwater of agricultural land in particular [12]. Nitrate contamination of groundwater has become a severe problem worldwide. Agricultural cultivation is the main factor causing nitrate contamination. High crop yields are achieved through the excessive use of chemical and organic fertilizers, and the long-term unsustainable application of nitrogen fertilizers has caused the accumulation of large amounts of residual nitrates in soil that readily leach into groundwater. Therefore, the nitrate contamination of groundwater in agricultural planting areas has become a pressing concern [13]. Higher concentrations of bioavailable P were also found in high-intensity agricultural systems [14]. Current strategies for reducing the loads and concentrations of P traveling from agricultural land to surface water have been the focus of many research papers [15]. Various legal instruments have regulated groundwater quality, and the evaluation of groundwater quality has been the main approach to ensure groundwater security and effective water management [16]. In agriculture, it is important to reduce the water scarcity in order to increase the crop yield, while stabilizing the environmental security by avoiding the direct pollution of rivers, canals, and surface water, conserving water and nutrients [17]. Wastewater treatment currently focuses on removing phosphorus (P) and nitrogen (N) due to their potential to cause eutrophication in bodies of water, as has been identified in surface water [18,19]. Farmland and pasture are considered to be the main contributors in the eutrophication of global coasts and oceans over the past century [20], and the nutrient level of agricultural exports is much higher than that of forests, grasslands, and other land types [21].

$NO_3$ pollution of the upper groundwater occurs almost everywhere in sandy and loamy soils [22]. Nitrate ($NO_3^-$) is a ubiquitous environmental pollutant that not only occurs naturally but is released by many human activities. These activities include the production and use of fertilizers, the combustion of fossil fuels (resulting in atmospheric deposition, hereafter AD), the leakage and discharge of both industrial and domestic sewage systems, and the alteration of natural vegetation with nitrogen (N) fixing crops [23]. The identification of nitrate ($NO_3^-$) sources and biogeochemical transformations is critical for understanding the different nitrogen (N) pathways, and thus for controlling diffuse pollution in groundwater affected by livestock and agricultural activities. A newly established internally constructed wetland was shown to be effective in treating diffuse agricultural pollution [24]. Diffuse agricultural pollution, especially from intensively managed agricultural lands, is a major cause of eutrophication; therefore, it is important to reduce the diffuse load to surface water. Constructed wetlands (CW) are an effective measure for improving water quality and reducing nutrient runoff from agriculture by using natural water treatment. Diffuse pollution is transferred from agricultural land to drainage ditches and larger water bodies, which causes the significant degradation of water quality in rivers and lakes [25]. Diffuse pollution is generally on a large scale, and its ambiguous nature makes it a challenge to address. Diffuse water pollution is a major problem in many agroecosystems, especially in irrigated areas linked to high-value ecosystems [6]. Pollution abatement policies often require the modification of agricultural practices, but they are usually rejected by farmers due to their impact on farm profitability, making point source pollution management an easier target for compliance [26].

Many countries have encouraged organic farming as an opportunity to reduce surface water pollution. Therefore, evaluating the relationship between water resource conservation and environmental protection investments, as well as establishing a reasonable model for ecological compensation, has become imperative [27]. The benefits of environmentally friendly practices may outweigh the costs when evaluating the environmental impact [28]. Since eutrophication is predicted to increase in the future due to population growth and climate change [29,30], it is critical to understand the ecological impacts of agricultural activities on aquatic ecosystems and implement management strategies for sustainable agricultural practices [26,31]. The aim of this research is to evaluate the impacts of organic and intensive agricultural activity on the water quality in four spots in the Nemunas River basin, Lithuania. Together with the implications on agricultural production type, we demonstrate the importance of considering both groundwater and surface water sampling, as only sampling surface water may hide the actual links between production type and water quality.

## 2. Materials and Methods

### 2.1. Study Area

To assess the impact of organic and intensive farming on the pollution of surface and groundwater, four organic and four intensive farms near rivers were selected.

The main criteria for the study site selection were as follows:

(a) The studied water bodies had to belong to the Nemunas River basin district (RBD) (the Dubysa, the Nevėžis, and the Neris small tributaries along with the Neris) and the Nemunas small tributaries along with the Nemunas basins);

(b) The distribution of the dominant landscape types in the selected basins had to be similar;

(c) A slope in the cultivated fields towards the rivers under study had to be present in all segregated and intensive farms;

(d) The development of crop production on all organic and intensive farms had to be present;

(e) In all organic and intensive farming farms, the soil must have a similar granulo-metric composition.

The basins of the studied water bodies are in the Panevėžys, Kėdainiai, Prienai, and Kaišiadorys Districts. The investigated river sections and the surface and groundwater sampling scheme are shown in Figures 1 and 2, respectively.

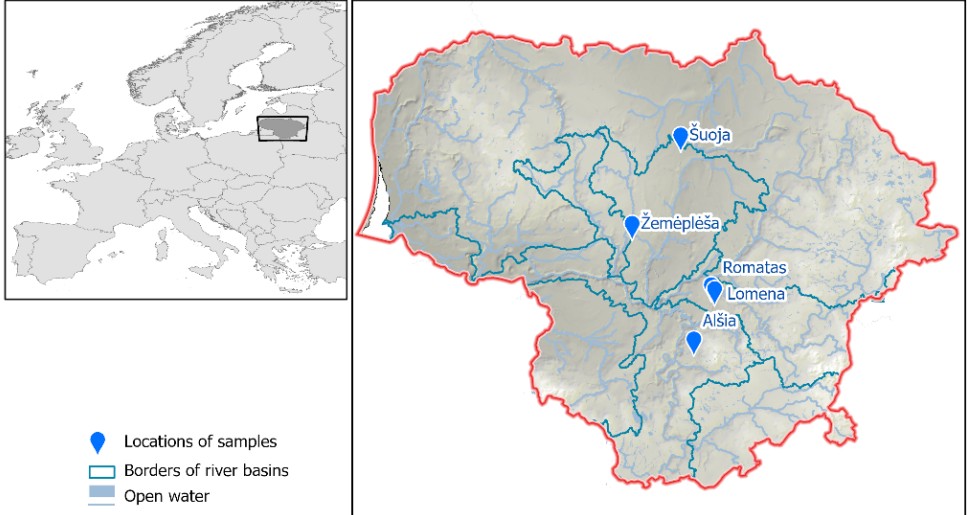

**Figure 1.** Location of study area and samples.

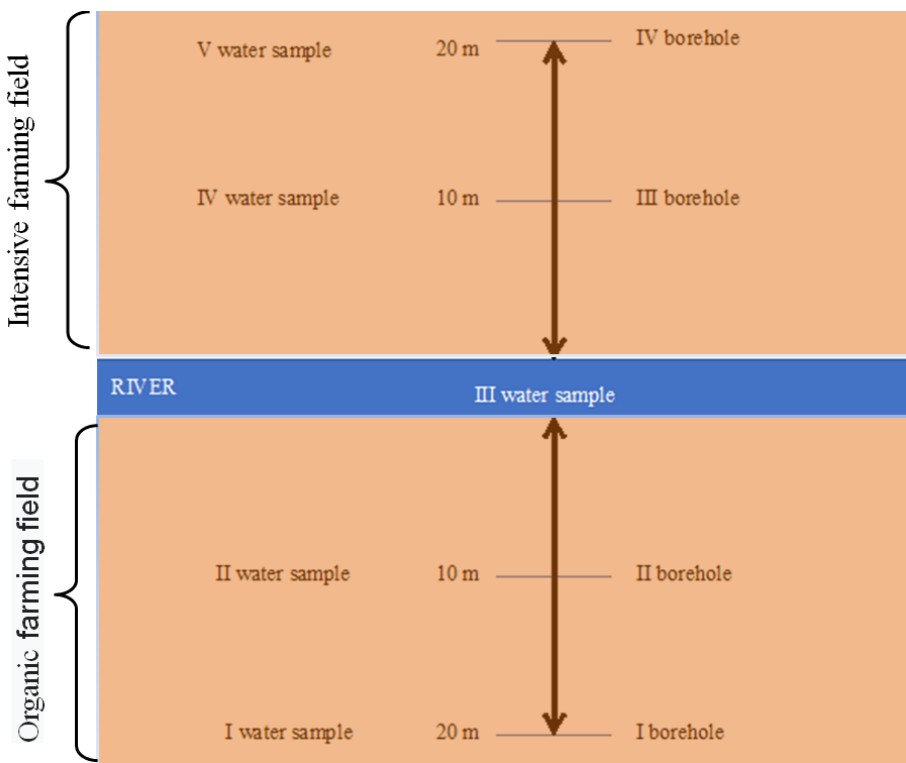

**Figure 2.** Surface and groundwater sampling scheme.

*2.2. Sample Preparation (Collection)*

Water research in five rivers located in the Nemunas River Basin District (Lomena and Romatas Kaišiadorys District, Žemėplėša Kėdainiai District, Šuoja Panevėžys District, and from Alšia River Prienai District) (Figure 1 and Table 1) was conducted every month from 2012 to 2016. During the study period, 108 surface water samples were collected from each river (a total of 324 samples were collected during the study period), and 23 wells 4–5 m deep for groundwater sampling (in the period 2014–2016) (Figure 2) were drilled in the cultivated fields near these rivers at 10 and 20 m perpendicular to the river shoreline. A total of 828 groundwater samples were collected from these wells during the study period.

**Table 1.** Hydrological characteristics of investigated rivers.

| River Name | Catchment Area, km$^2$ | Length, km | Average Flow Rate, m$^3$/s |
|---|---|---|---|
| Alšia | 137.8 | 40.70 | 0.89 |
| Lomena | 186.9 | 36.22 | 1.14 |
| Romatas | 123.3 | 5.12 | 0.05 |
| Šuoja | 245 | 47.05 | 1.19 |
| Žemėplėša | 8.8 | 5.64 | 0.04 |

Surface and groundwater samples were collected and transported using 1.5 L plastic containers with screw caps.

Surface water samples were collected upstream after rinsing the sampling vessel with surface water. The plastic container collected samples approximately 0.5 m from each riverbank, and the container was closed immediately after sampling.

Groundwater samples were pumped from the wells into plastic 1.5 L containers with a rubber tube. The samples were transported to the Vytautas Magnus University Agricultural Academy Environmental Laboratory, where the quality of surface and groundwater was assessed according to the indicators that best describe the quality of river water.

The surface water quality was assessed according to the following parameters: nitrate nitrogen ($NO_3$-N) (LST ISO 10304-1: 2009), nitrites ($NO_2$-) (LST EN 26777: 1999), ammonium nitrogen ($NH_4$-N) (LST ISO 7150-1: 1998), total nitrogen ($N_t$) (LST EN ISO 11905-1: 2000), orthophosphate phosphorus ($PO_4$-P) (spectrophotometric), total phosphorus ($P_t$) (LST EN ISO 6878: 2004), and hydrogen ion concentration (pH) (LST EN ISO 10523: 2012). The ecological status of rivers was determined according to the indicators of physicochemical quality elements, using the methodology for determining the status of surface water bodies.

The groundwater quality was assessed according to the following parameters: nitrates ($NO_3^-$) (LST ISO 10304-1: 2009), nitrites ($NO_2^-$) (LST EN 26777: 1999), ammonium ions ($NH_4^+$) (LST ISO 7150-1: 1998), total nitrogen ($N_t$) (LST EN ISO 11905-1: 2000), orthophosphates ($PO_4^{+3}$) (spectrophotometric), total phosphorus ($P_t$) (LST EN ISO 6878: 2004), and hydrogen ion concentration (pH) (LST EN ISO 10523: 2012).

### 2.3. Statistical Analyses

Differences between the values of water quality indicators determined in the areas of organic and intensive agricultural production were assessed by calculating the criteria *t*-test between the independent values. The difference was statistically significant at $p \leq 0.05$. The obtained research data were processed using STATISTICA 10 (Palo Alto, CA, USA).

## 3. Results

### 3.1. Ecological Statuses of Rivers Stretch According to Physicochemical Quality Elements Values

To assess the ecological status of surface water, water samples were collected from five rivers in the Nemunas River Basin District (Lomena, Romatas, Žemėplėša, Šuoja, and Alšia). The ecological status was assessed by total P, total N, $NO_3$-N, $NH_4$-N, and $PO_4$-P. The results are shown in Figure 3.

The ecological status classes according to methodology for determining the condition of surface water bodies [32] are --- Very good; --- Good; --- Average; --- Bad; and --- Very bad.

According to the concentration of total phosphorus in the water, the water quality of the majority of rivers (Romatas, Lomena, and Šuoja Rivers) was very good throughout all the studied years. The results for the Alšia River were good in 2012–2013 and very good in 2014–2016. The results for the Žemėplėša River were average in 2012, good from 2014 to 2016, and in 2013 and 2015, it had a very good ecological status. According to the concentrations of orthophosphates and ammonium ions determined in 2012–2016, the water quality of all studied rivers had a very good ecological status.

Judging by the concentrations of total nitrogen, the ecological status in the Alšia River was average in 2012 and in 2013–2016, the ecological status was good. The water quality of the Romatas River in 2012, 2014, and 2016 was very good, average in 2013, and in 2015, the results were good. The water quality of the Lomena River in 2013–2016 was average, whereas in 2012, the ecological status was rather poor. The Šuoja River water quality was good in 2012, and in 2013, the results were only average. Moreover, from 2014 to 2016, the water quality of the Šuoja River was the worst, as compared to all other rivers studied during the same time periods. In 2012–2013 and 2015, the water quality of the Žemėplėša River was average, and in 2014 and 2016, the results were poor.

For nitrate concentration, the water quality of the Alšia and Romatas Rivers was very good, and the water quality of the Lomena, Šuoja, and Žemėplėša Rivers was of medium ecological status in almost all the study years (the ecological status of the Žemėplėša River in 2014 and the Šuoja River in 2012 was poor).

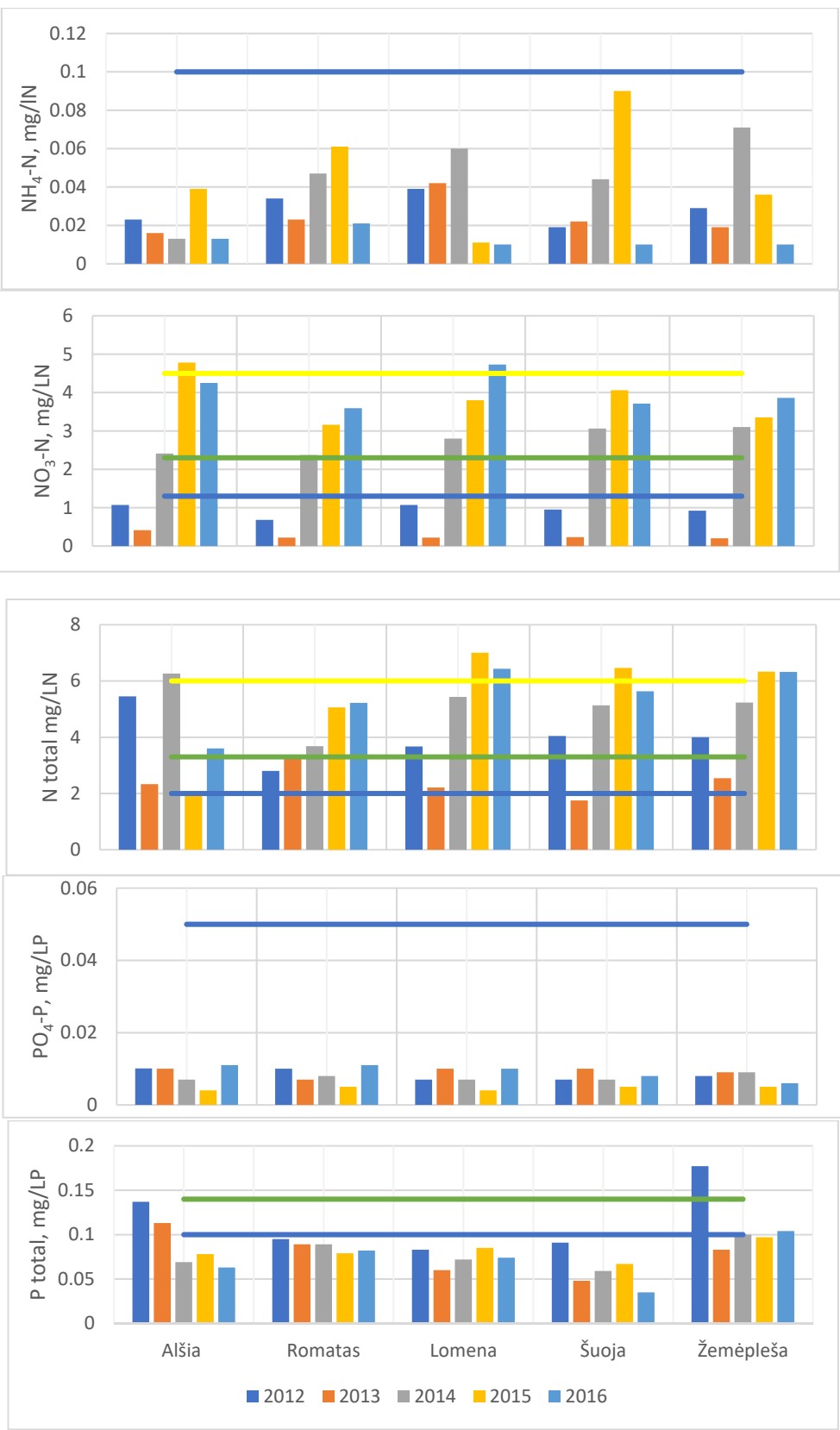

**Figure 3.** Ecological status of rivers according to the indicators of the physicochemical quality of elements.

### 3.2. Comparative Analysis of Nutrient Dispersion in Water in Organic and Conventional Farming Areas

To assess the distribution of nutrients from organic and intensive agricultural activities, groundwater was collected from the drilled wells, and we analyzed the total P, the total N, $NO_3$-N, and the concentrations of $NH_4$-N and $PO_4$-P. The total P concentration scatter plots are shown in Figure 4, together with the relevant properties of the surface water.

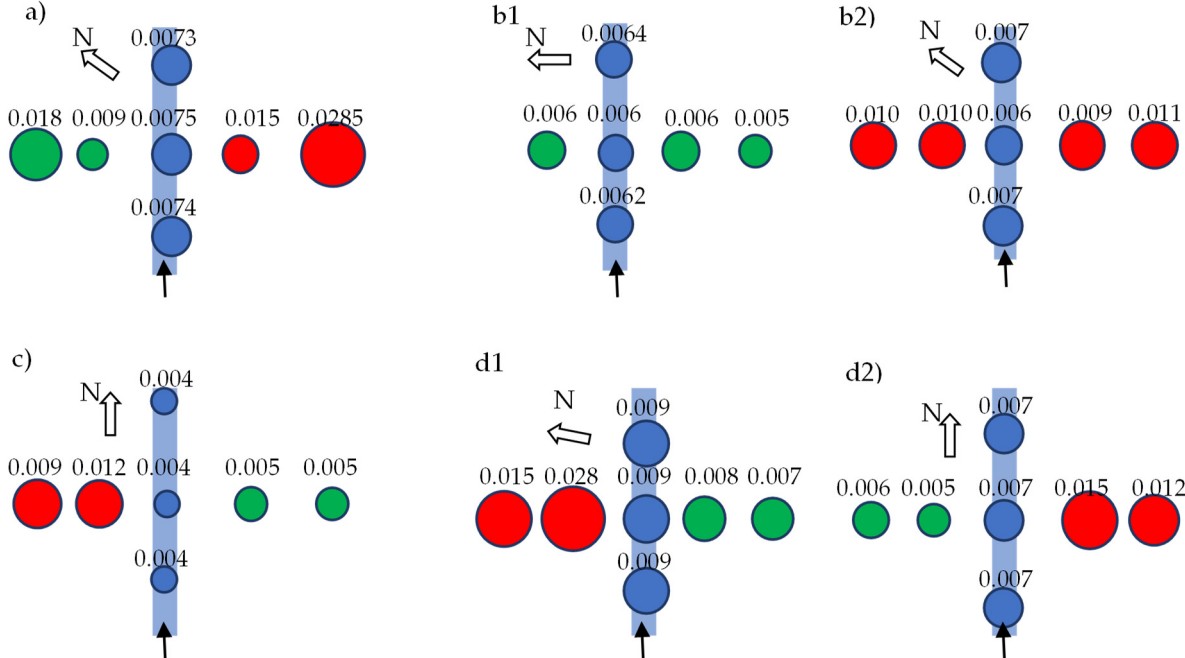

**Figure 4.** Concentrations of total phosphorus (total P) in groundwater of rivers and boreholes. Blue dots refer to the surface water samples, green dots refer to the ground water samples taken in organic farming areas and red dots refer to the ground water samples taken in intensive farming areas: (**a**) Alšia, (**b1**) and (**b2**) Žemėplėša, (**c**) Šuoja, (**d1**) Romatas and (**d2**) Lomena.

Assessing the spread of pollution near the Žemėplėša River in Kėdainiai District, the concentrations of $P_t$ in the groundwater of the boreholes drilled in organic farming areas varied from 0.059 to 0.06 mg/L, and in the areas of intensive farming, the concentrations of $P_t$ varied from 0.089 to 0.091 mg/L. The concentrations of $P_t$ in the groundwater of boreholes drilled at 20 m from the river were higher, as compared to the concentrations in boreholes drilled at 10 m. Concentrations closer to the river were lower.

Near the Romatas River in the Kaišiadorys District, the $P_t$ concentrations in the groundwater after drilling in areas of intensive agriculture were higher, at 0.144 and 0.147 mg/L. The concentrations of $P_t$ in the groundwater of boreholes drilled at 20 m from the river were lower, as compared to the concentrations in boreholes drilled at 10 m. The concentrations closer to the river were higher.

Near the Romatas River in the Kaišiadorys District, the $P_t$ concentrations in the groundwater of the boreholes drilled in ecological agriculture land areas were 0.056 and 0.052 mg/L, and in intensive agriculture areas, the $P_t$ concentrations were higher, at 0.119 and 0.118 mg/L. The concentrations of $P_t$ in the groundwater of boreholes, drilled at 20 m from the river were higher, as compared to the concentrations in boreholes drilled at 10 m. The concentrations closer to the river decreased.

Near the Alšia River in the Prienai District, the $P_t$ concentrations in the groundwater of the boreholes drilled in organic farming areas were 0.169 and 0.198 mg/L, and in intensive farming areas, the $P_t$ concentrations were higher, at 0.266 and 0.267 mg/L. The concentrations of $P_t$ in the groundwater drilled at 20 m from the river were lower, as

compared to the concentrations in the boreholes drilled at 10 m. The concentrations closer to the river increased.

Near the Šuoja River in Panevėžys District, the $P_t$ concentrations in the groundwater of the boreholes drilled in organic farming areas were 0.059 and 0.060 mg/L, and in intensive farming areas, the $P_t$ concentrations were higher, at 0.091 and 0.089 mg/L. The concentrations of $P_t$ in the groundwater of boreholes drilled at 20 m from the river in organic farming areas were lower compared to the concentrations in boreholes drilled at 10 m from the river. The concentrations closer to the river increased, and in areas of intensive agriculture, they decreased. The terrain affected the current situation.

The total phosphorus content of all groundwater samples collected from boreholes drilled on organic farms was lower than that in groundwater samples collected from boreholes drilled on intensive farms.

The concentrations of total phosphorus in rivers were lower than in the groundwater samples collected from boreholes drilled in intensive farming areas, but higher or very similar to the concentrations in water samples collected from boreholes drilled in organic farming areas (except in Prienai District, where the concentrations of phosphorus in the river were lower).

The total N concentration scatter plots are shown in Figure 5.

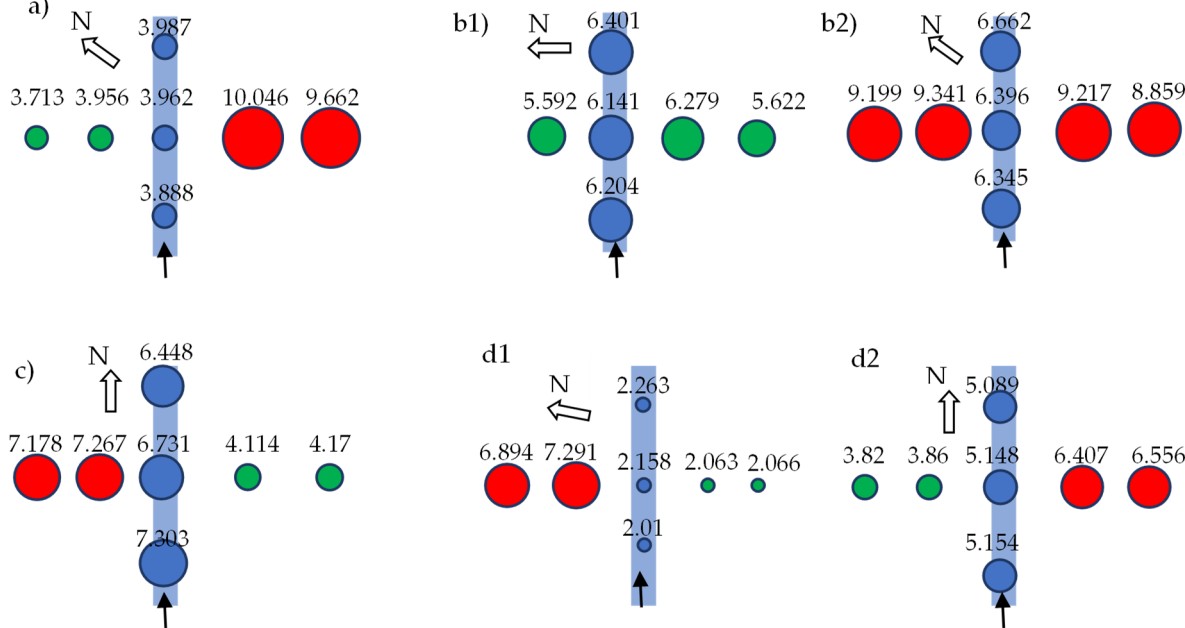

**Figure 5.** Concentrations of total nitrogen ($N_t$) in the groundwater of rivers and boreholes. Blue dots refer to the surface water samples, green dots refer to the ground water samples taken in organic farming areas and red dots refer to the ground water samples taken in intensive farming areas: (**a**) Alšia, (**b1**) and (**b2**) Žemėplėša, (**c**) Šuoja, (**d1**) Romatas and (**d2**) Lomena.

Assessing the spread of pollution near the Žemėplėša River in Kėdainiai District, the concentrations of Nt in the groundwater of the boreholes drilled in the areas of organic farming varied from 5.592 to 6.279 mg/L. The concentrations of $N_t$ in the groundwater of boreholes drilled 20 m from the river were lower, as compared to the concentrations in boreholes drilled at 10 m. The concentrations closer to the river increased.

Near the Lomena River in Kaišiadorys District, the concentrations of $N_t$ in the groundwater of the boreholes drilled in the areas of organic farming were 3.82 and 3.86 mg/L, and in the areas of intensive farming, the concentrations of $N_t$ were 6.407 and 6.566 mg/L. The concentrations of $N_t$ in the groundwater of the boreholes drilled in organic farming at 20 m from the river were lower, as compared to the concentrations in boreholes drilled at 10 m from the river. Meanwhile, the concentrations of $N_t$ in the groundwater of the wells

drilled in intensive agricultural land at 20 m from the river were higher, as compared to the concentrations in the wells drilled at 10 m. This indicated that the concentrations of $N_t$ decreased closer to the river.

Near the Romatas River in Kaišiadorys District, the $N_t$ concentrations in the groundwater of the boreholes drilled in organic farming land areas were 2.063 and 2.066 mg/L, and in intensive farming areas, the $N_t$ concentrations were 6.894 and 7.291 mg/L. The concentrations of $N_t$ in the groundwater of the boreholes drilled in organic farming at 20 m from the river were higher, as compared to the concentrations in boreholes drilled at 10 m. The concentrations decreased closer to the river. Meanwhile, the concentrations of $N_t$ in the groundwater of the boreholes drilled in intensive agricultural land at 20 m from the river were lower, as compared to the concentrations in the boreholes drilled at 10 m. This indicated that the concentrations of $N_t$ increased closer to the river. The terrain had affected the outcome.

Near the Alšia River in Prienai District, the $N_t$ concentrations in the groundwater of the boreholes drilled in organic farming land areas were 3.713 and 3.956 mg/L, and in intensive farming areas, the $N_t$ concentrations were higher, at 9.662 and 10.046 mg/L. The concentrations of $N_t$ in the groundwater of boreholes drilled at 20 m from the river were lower, as compared to the concentrations in boreholes drilled at 10 m. The concentrations increased closer to the river.

Near the Šuoja River in Panevėžys District, the $N_t$ concentrations in the groundwater of the boreholes drilled in organic farming land areas were 4.114 and 4.17 mg/L in intensive farming areas, the $N_t$ concentrations were higher, at 7.178 and 7.267 mg/L. The concentrations of $N_t$ in the groundwater of wells drilled in organic farming areas at 20 m from the river were higher compared to the concentrations in boreholes drilled at 10 m. The concentrations decreased closer to the river, and in areas of intensive agriculture, they increased. The terrain had affected the outcome.

The total nitrogen content of all the groundwater samples collected from boreholes drilled on organic farms was lower than that of the groundwater samples collected from boreholes drilled on intensive farms.

The concentrations of total nitrogen in the rivers were lower than in the groundwater samples collected from the boreholes drilled in the areas of intensive farms, but higher or very similar to the concentrations in the water samples collected from the boreholes drilled in organic farms.

The dispersion diagrams of the $NO_3$-N concentrations are shown in Figure 6.

Near the Žemėplėša River in Kėdainiai District, the concentrations of $NO_3$-N in the groundwater of the boreholes drilled in the areas of organic farming land and ecological agricultural rivers varied from 2.213 to 2.275 mg/L. The $NO_3$-N concentrations in the groundwater of the boreholes drilled 20 m upstream were lower or very similar to the concentrations in the boreholes drilled at 10 m. The concentrations increased closer to the river.

Near the Lomena River in Kaišiadorys District, the concentrations of $NO_3$-N in the groundwater of the boreholes drilled in the areas of intensive farming were higher, at 3.768 and 3.751 mg/L. The concentrations of $NO_3$-N in the groundwater of wells drilled in organic farming at 20 m from the river were higher, as compared to the concentrations in boreholes drilled at 10 m from the river. The concentrations decreased closer to the river. Meanwhile, the $NO_3$-N concentrations in the groundwater of wells drilled in intensive agricultural land at 20 m from the river were lower, as compared to the concentration in wells drilled at 10 m. This indicated that $NO_3$-N concentrations increased closer to the river. The terrain had an effect.

The concentrations of $NO_3$-N in the groundwater of the boreholes drilled in organic agricultural land near the Romatas River in Kaišiadorys District were 0.219 and 0.223 mg/L, and in intensive agricultural areas, higher $NO_3$-N concentrations were found, at 0.892 and 0.917 mg/L. The concentrations of $NO_3$-N in the groundwater of wells drilled at 20 m from

the river were higher, as compared to the concentrations in boreholes drilled at 10 m. The concentrations decreased closer to the river.

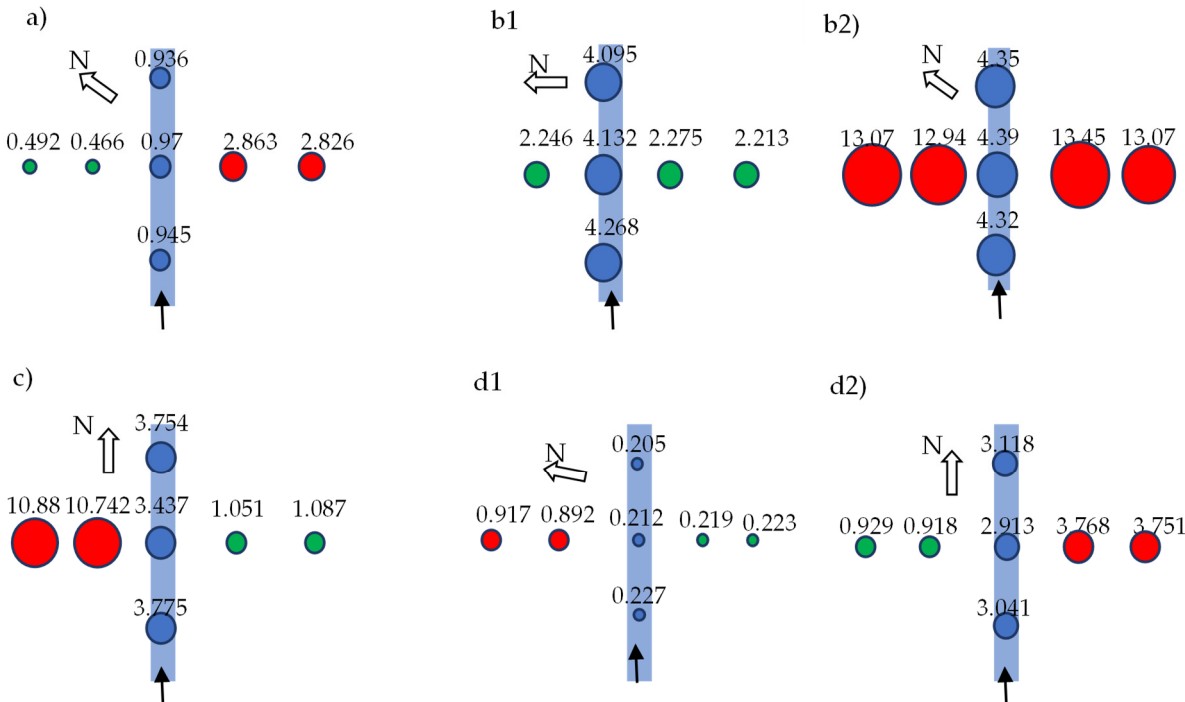

**Figure 6.** Concentrations of nitrate nitrogen ($NO_3$-N) in the groundwater of rivers and boreholes. Blue dots refer to the surface water samples, green dots refer to the ground water samples taken in organic farming areas and red dots refer to the ground water samples taken in intensive farming areas: (**a**) Alšia, (**b1**) and (**b2**) Žemėplėša, (**c**) Šuoja, (**d1**) Romatas and (**d2**) Lomena.

Near the Alšia River in Prienai District, the $NO_3$-N concentrations in the groundwater of the boreholes drilled in organic farming land areas were 0.466 and 0.492 mg/L, and in intensive farming, the $NO_3$-N concentrations were 2.863 and 2.826 mg/L. The concentrations of $NO_3$-N in the groundwater of wells drilled in organic farming areas at 20 m from the river were higher, as compared to the concentrations in wells drilled at 10 m. The concentrations decreased closer to the river, whereas in areas of intensive agriculture, they increased.

Near the Šuoja River in Panevėžys District, the $NO_3$-N concentrations in the groundwater of the boreholes drilled in organic farming land areas were 1.051 and 1.087 mg/L, and in intensive farming areas, the $NO_3$-N concentrations were higher, at 10.742 and 10.887 mg/L. The concentrations of $NO_3$-N in the groundwater of wells drilled at 20 from the river were higher, as compared to the concentrations in boreholes drilled at 10 m. The concentrations decreased closer to the river.

All groundwater samples collected from boreholes drilled on organic farms had lower $NO_3$-N levels than the groundwater samples collected from boreholes drilled on intensive farms.

The $NO_3$-N concentrations in the rivers were lower than in the groundwater samples collected from the boreholes drilled in the intensive farming areas, but higher or very similar to the concentrations found in the water samples collected from the boreholes drilled in organic farms.

The dispersion diagrams of the $NH_4$-N concentration are shown in Figure 7.

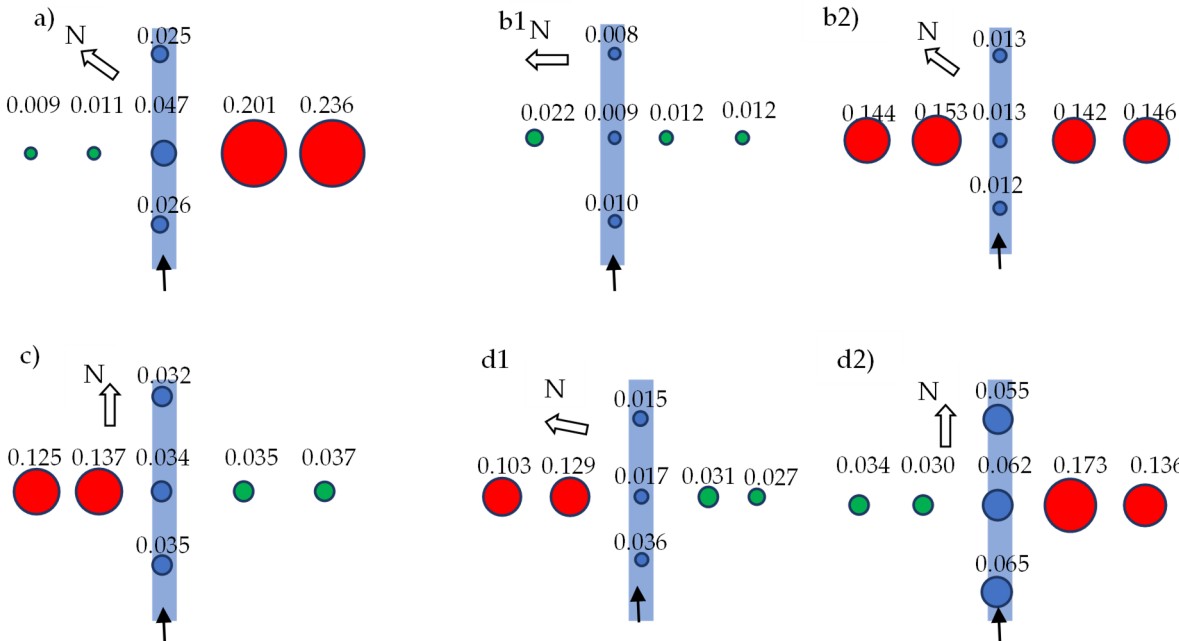

**Figure 7.** Concentrations of ammonium nitrogen (NH₄-N) in the groundwater of rivers and boreholes. Blue dots refer to the surface water samples, green dots refer to the ground water samples taken in organic farming areas and red dots refer to the ground water samples taken in intensive farming areas: (**a**) Alšia, (**b1**) and (**b2**) Žemėplėša, (**c**) Šuoja, (**d1**) Romatas and (**d2**) Lomena.

Near the Žemėplėša River in Kėdainiai District, the concentrations of NH₄-N in the groundwater of the boreholes drilled in the areas of organic farming land were 0.0122–0.0229 mg/L, and in the areas of intensive agriculture, it was higher, at 0.1425–0.153 mg/L. The concentrations of NH₄-N in the groundwater of the wells drilled at 20 m upstream were lower or very similar to the concentrations in the wells drilled at 10 m. The concentrations increased closer to the river.

Near the Lomena River in Kaišiadorys District, the NH₄-N concentrations in the groundwater from boreholes drilled in the areas of organic farming were 0.0305 and 0.0348 mg/L, and in intensive farming areas, the NH₄-N concentrations were higher, at 0.1337 and 0.1731 mg/L. The concentrations of NH₄-N in the groundwater of the boreholes drilled in organic farming at 20 m from the river were higher, as compared to the concentrations in boreholes drilled at 10 m. The concentrations decreased closer to the river. Meanwhile, the concentrations of NH₄-N in the groundwater of wells drilled in intensive agricultural land at 20 m from the river were lower, as compared to the concentrations in boreholes drilled at 10 m upstream. This indicated that the concentrations of NH₄-N increased closer to the river.

Near the Romatas River in Kaišiadorys District, the NH₄-N concentrations in the groundwater from boreholes drilled in ecological agriculture land areas were 0.0273 and 0.0312 mg/L, and in intensive agriculture areas, the NH₄-N concentrations were higher, at 0.1298 and 0.1037 mg/L. The concentrations of NH₄-N in the groundwater of wells drilled at 20 m from the river were lower, as compared to the concentrations in wells drilled at 10 m. The concentrations increased closer to the river.

Near the Alšia River in Prienai District, the NH₄-N concentrations in the groundwater of the boreholes drilled in organic farming areas were 0.011 and 0.0094 mg/L, and in intensive farming areas, the NH₄-N concentrations were higher, at 0.201 and 0.2369 mg/L. The concentrations of NH₄-N in the groundwater of boreholes drilled at 20 m from the river in organic farming areas were lower, as compared to the concentrations in boreholes drilled at 10 m from the river. The concentrations increased closer to the river, whereas in areas of intensive agriculture, it decreased.

Near the Šuoja River in Panevėžys District, the $NH_4$-N concentrations in the groundwater of the boreholes drilled in organic farming areas were 0.0351 and 0.0375 mg/L, and in intensive farming areas, the $NH_4$-N concentrations were higher, at 0.1257 and 0.137 mg/L. The concentrations of $NH_4$-N in the groundwater of boreholes drilled at 20 m from the river in organic farming areas were higher, as compared to the concentrations in boreholes drilled at 10 m. The concentrations decreased closer to the river, whereas in areas of intensive agriculture, it increased.

All groundwater samples collected from boreholes drilled on organic farms had lower $NH_4$-N levels than groundwater samples collected from boreholes drilled on intensive farms.

The concentrations of $NH_4$-N in rivers were lower than in the groundwater samples collected from boreholes drilled in intensive farming areas, but higher or very similar to those found in water samples collected from boreholes drilled in organic farming areas.

The dispersion diagrams of $PO_4$-P concentrations are shown in Figure 8.

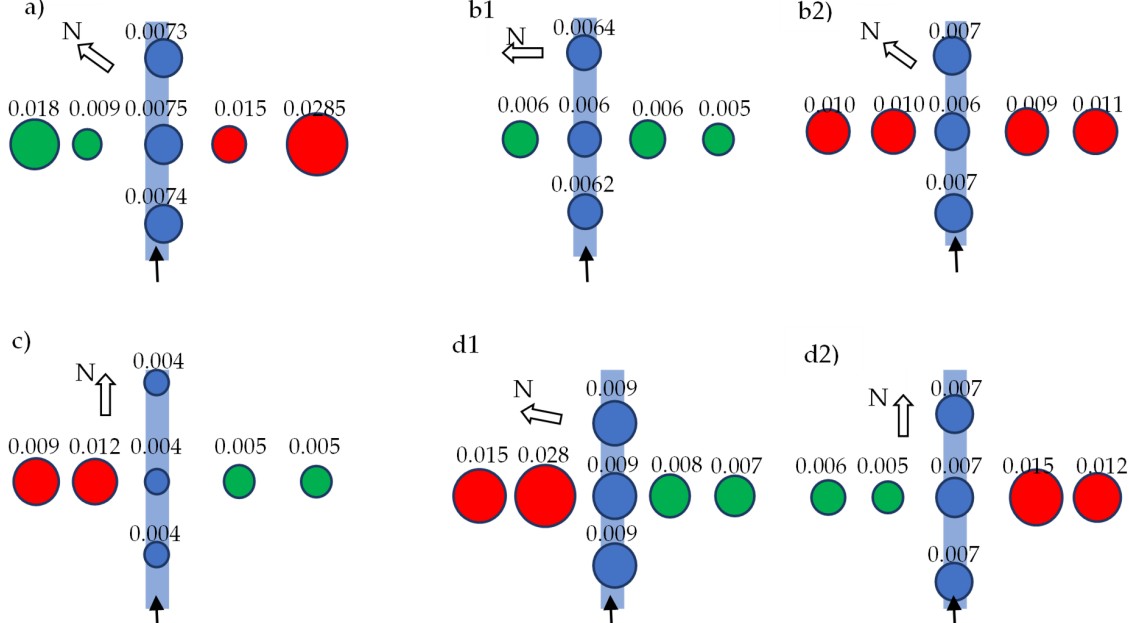

**Figure 8.** Phosphate phosphorus ($PO_4$-P) concentrations in the groundwater of rivers and boreholes. Blue dots refer to the surface water samples, green dots refer to the ground water samples taken in organic farming areas and red dots refer to the ground water samples taken in intensive farming areas: (**a**) Alšia, (**b1**) and (**b2**) Žemėplėša, (**c**) Šuoja, (**d1**) Romatas and (**d2**) Lomena.

Assessing the spread of pollution near the Žemėplėša River in Kėdainiai District, the concentrations of $PO_4$-P in the groundwater of the boreholes drilled in the areas of organic farming varied from 0.00058 to 0.0061 mg/L, and in areas of intensive agriculture, the $PO_4$-P concentrations were higher, ranging from 0.0092 to 0.0111 mg/L. The concentrations of $PO_4$-P in the groundwater drilled at 20 m from the river in organic farming areas were lower compared to the concentrations in boreholes drilled at 10 m. The concentrations increased closer to the river, whereas in areas of intensive agriculture, they decreased.

Near the Lomena River in Kaišiadorys District, the concentrations of $PO_4$-P in the groundwater of wells drilled in the areas of organic farming were 0.0056 and 0.006 mg/L, and in the areas of intensive farming, the concentrations of $PO_4$-P were 0.0155 and 0.0126 mg/L. The concentrations of $PO_4$-P in the groundwater of the boreholes drilled in organic farming at 20 m from the river were higher compared to the concentrations in boreholes drilled at 10 m. The concentrations decreased closer to the river. Meanwhile, the concentrations of $PO_4$-P in the groundwater of the boreholes drilled in intensive agricultural land at 20 m from the river were lower, when compared to the concentrations in boreholes drilled at 10 m from the river. This indicated that the $PO_4$-P concentrations increased closer to the river.

The concentrations of $PO_4$-P in the groundwater of the boreholes drilled in the areas of organic agriculture near the Romatas River in Kaišiadorys District were 0.0082 and 0.0079 mg/L, and in areas of intensive agriculture, the concentrations of $PO_4$-P were 0.0285 and 0.0158 mg/L. The concentrations of $PO_4$-P in the groundwater drilled at 20 m from the river were lower compared to the concentrations in the boreholes drilled at 10 m. The concentrations increased closer to the river.

Near the Alšia River in Prienai District, the concentrations of $PO_4$-P in the groundwater of the boreholes drilled in organic farming areas were 0.0212 and 0.0181 mg/L, and in intensive farming areas, the concentration of $PO_4$-P were 0.0057 and 0.0.0285 mg/L. The concentrations of $PO_4$-P in the groundwater drilled at 20 m from the river in organic farming areas were lower compared to the concentrations in boreholes drilled at 10 m. The concentrations increased closer to the river, whereas in areas of intensive agriculture, they decreased.

Near the Šuoja River in Panevėžys District, the concentrations of $PO_4$-P in the groundwater of the boreholes drilled in organic farming areas were 0.0059 and 0.0057 mg/L, and in intensive farming areas, the $PO_4$-P concentrations were higher, at 0.0127 and 0.0096 mg/L. The concentrations of $PO_4$-P in the groundwater drilled at 20 m from the river were lower compared to the concentrations in the boreholes drilled at 10 m. The concentrations increased closer to the river.

In all groundwater samples (except in Prienai District) collected from boreholes drilled in organic farming areas, the amount of $PO_4$-P was lower than in the groundwater samples collected from boreholes drilled in intensive farming areas.

The concentrations of $PO_4$-P in the rivers were lower than in the groundwater samples collected from the boreholes drilled in the areas of intensive farms, but higher or very similar to the concentrations found in the water samples collected from the boreholes drilled in the areas of organic farms.

### 3.3. Comparative Analysis of Nutrient Concentrations in Water in Organic and Conventional Farming Areas

To compare pH, the $P_t$, $N_t$, $NO_3$-N, $NH_4$-N and $PO_4$-Pconcentrations in the groundwater of the boreholes drilled in organic and intensive farming areas were calculated as average values. Differences between the values of water quality indicators determined in the areas of organic and intensive agricultural production were assessed by calculating the criteria of a *t*-test between the independent values. The difference was statistically significant at $p \leq 0.05$. The results are shown in Table 2.

**Table 2.** Differences between the values of water quality indicators in the areas of organic and intensive agricultural production.

| Water Quality Indicator | Kėdainiai District | Prienai District | Kaišiadorys District | Panevėžys District |
|---|---|---|---|---|
| pH | *t*-test = 7.051 <br> *p* = 0.003 | *t*-test = 3.996 <br> *p* = 0.009 | *t*-test = 0.213 <br> *p* = 0.83 | *t*-test = 3.170 <br> *p* = 0.045 |
| $PO_4$-P, mg/L P | *t*-test = 4.798 <br> *p* = 0.0001 | *t*-test = 0.484 <br> *p* = 0.64 | *t*-test = 5.654 <br> *p* = 0.0001 | *t*-test = 6.94 <br> *p* = 0.00004 |
| $NH_4$-N mg/L N | *t*-test = 22.752 <br> *p* = 0.00000 | *t*-test = 18.454 <br> *p* = 0.0000 | *t*-test = 10.622 <br> *p* = 0.0001 | *t*-test = 11.680 <br> *p* = 0.00001 |
| $NO_3$-N, mg/L N | *t*-test = 45.899 <br> *p* = 0.00000 | *t*-test = 56.347 <br> *p* = 0.0000 | *t*-test = 3.913 <br> *p* = 0.0001 | *t*-test = 5.084 <br> *p* = 0.0005 |
| N total, mg/L N | *t*-test = 9.492 <br> *p* = 0.0000 | *t*-test = 22.726 <br> *p* = 0.0000 | *t*-test = 12.212 <br> *p* = 0.0000 | *t*-test = 12.191 <br> *p* = 0.0000 |
| P total mg/L P | *t*-test = 22.952 <br> *p* = 0.0000 | *t*-test = 6.476 <br> *p* = 0.0000 | *t*-test = 9.295 <br> *p* = 0.0000 | *t*-test = 8.252 <br> *p* = 0.0000 |

Considering the total concentrations, the concentrations of pH, $PO_4$-P, $NH_4$-N, $NO_3$-N, $N_t$, and $P_t$ in the groundwater of all boreholes drilled in areas of organic farming were lower than in the groundwater from boreholes drilled in the areas of intensive farming. The differences are statistically significant, except for the differences between pH values in Kaišiadoriai district and $PO_4$-*p* values in Prienai district.

## 4. Discussion

In this study, we found that all rivers had a good or very good ecological status, according to the $PO_4$-P and $NH_4$-N concentrations in surface water samples taken from organic and intensive farming areas. According to its $P_t$ values, only one river did not correspond to the values of good or very good ecological status. However, according to total N, four rivers did not have a good or very good ecological status. The knowledge of the ecological status of rivers is extremely important, as high nitrate concentrations in water bodies cause numerous ecological problems, such as algal blooms [33], the eutrophication of lakes and reservoirs, and the eradication of species in river ecosystems [34–36]. Therefore, studies of water quality issues have gained attention in recent years. It is obvious that knowledge of water quality is required to manage the ecological status of water resources. The increased rate of water resource use has resulted in their contamination by the wastewater of domestic, industrial, and agricultural sectors. As previously mentioned, agricultural pollution poses a considerable challenge to crop security and human health, especially in economically developed areas [4]. However, our study has identified that some factors determining water quality may be hidden if focusing on only surface water sampling. There are numerous studies evaluating the strategies for reducing P loads and concentrations from agricultural land to surface water. Usually, they suggest several measures to control the transport of agricultural P from soil to water, such as optimizing the efficiency of fertilizer P use, refining animal feed rations, using feed additives, increasing animal P uptake by moving manure from surplus to deficit sites and applying conservation practices such as reduced tillage, buffer strips, and covering crops in areas of critical P export from the watershed [15]. It is commonly accepted that assessing surface water quality is often complex, expensive, and tedious, as well as an impractical task. Use of modelling-based analyses that consider all biophysical and chemical parameters and provide a consistent assessment may be considered [37,38]; however, this has not been considered in our study yet.

Assessing the nutrient dispersion from organic and intensive farming activities, we found that almost all groundwater samples collected from organic farm wells had a lower nutrient content than the groundwater samples collected from intensive farming wells.

By comparing the values of pH, total P, total N, and $NO_3$-N, as well as the concentrations of $NH_4$-N and $PO_4$-P, in groundwater samples taken from boreholes drilled in ecological and intensive agricultural areas, we found that, overall, they were lower in organic farming areas compared to intensive farming areas. The decrease in total phosphorus and nitrogen concentration in water due to the increase in the area of organic farms occurs due to the balancing of the use of nitrogen (N), phosphorus (P), and potassium (K) preparations in organic farms [39]. Such research results could have been influenced by the fact that in the research period in intensive agricultural farms, the largest areas of cultivated land were occupied by grain crops (from 75 to 86%). However, the majority of crops in ecological agriculture farms focus on cultivation of perennial grasses, which ensure a higher absorption of nutrients and less nutrient material loss through drainage systems compared to the areas where cereals or other crops are grown [39–41]. Intensive agricultural use of large amounts of inorganic fertilizers, together with the return flow associated with irrigation, increases the concentration of $NO_3$ in surface water bodies and especially in the groundwater of agricultural areas [12]. Many countries have encouraged organic farming as an opportunity to reduce surface water pollution. Therefore, evaluating the relationship between water resource conservation and environmental protection investments, as well as establishing a reasonable model for ecological compensation, has become imperative [27]. The benefits of environmentally friendly practices may outweigh

the costs when evaluating the environmental impact [28]. Since eutrophication is predicted to increase in the future due to population growth and climate change [29,30], it is critical to understand the ecological impacts of agricultural activities on aquatic ecosystems and implement management strategies for sustainable agricultural practices [26,31].

## 5. Conclusions

All five studied rivers presented an at least moderate ecological status according to the quality of surface water samples, which did not change during the period of observation. However, the quality of ground water sampled in proximity to the water streams investigated was strongly dependent on the type of farming in the fields the samples were taken from, i.e., the concentrations of $P_t$, $N_t$, $NO_3$-N, $NH_4$-N, and $PO_4$-P were statistically significantly lower in almost in all samples taken in organic farming fields than the ones collected from the boreholes drilled in intensive agricultural farms. The concentrations of $Pt$, $Nt$, $NO_3$-N, $NH_4$-N, and $PO_4$-P in surface water samples exceeded or were equal to the ground water concentration in organic farming fields, but they were lower than the concentrations in ground water samples in intensive agricultural farms. This suggests the importance of ground water sampling to understand the factors influencing the water quality. Furthermore, this also confirms the assumption that the leaching of nutrients into underground and surface water in the areas of organic agricultural production is statistically significantly lower than in intensive agriculture farms. Therefore, in order to improve the quality of water, it is extremely important to increase both the area of organic farms and the proportion of organic farms, say, to at least 25%, as assumed in EU recommendations.

We accept that there are numerous research challenges remaining for future studies to deepen the understanding of farming impacts on the quality of both ground and surface water. Extensive field sampling needs to be coupled with modelling exercises. Sampling schemes need to be further tested to better understand the spatial patterns of the impacts. It is extremely important to assess the seasonal impacts on these processes, e.g., the nitrogen concentrations in the soil need to be assessed in autumn, as nitrogen may migrate down through the soil and pollute the ground water in warm winters when no winter freeze occurs.

**Author Contributions:** L.Č. (Laima Česonienė) and K.T. designed the study and L.Č. (Laura Čiteikė), G.M. and D.Š. performed the experiments, analyzed the data, and wrote the manuscript. All authors have read and agreed to the published version of the manuscript.

**Funding:** This research was funded by LMT grant number S-LJB-20-3.

**Data Availability Statement:** The data are available in a publicly accessible repository.

**Conflicts of Interest:** The authors declare no conflict of interest.

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
