# Peer review of "The Impact of Organic and Intensive Agricultural Activity on Groundwater and Surface Water Quality"

_water, doi:10.3390/w15061240_

Round 1
Reviewer 1 Report (Previous Reviewer 3)
In spite of the corrections, discussion chapter still remains the week point of this manuscript and the authors did not took in consideration my previous remarks. In fact, as I mentioned before, the discussion lacks a convenient analysis of the chemical and physical processes that retain nutrients. Besides, nothing is mentioned about the techniques that can be used in intensive farming to reduce the eutrophication of superficial waters (replacement by organic farming is hardly possible in many situations), which could represent a significant contribution of this paper to the appropriate management and rehabilitation of streams impacted by intensive farming. Aspects like the type of soils, geology, riparian vegetation and slope/topography or the potential impact in the drainage from artificialization of the riverine areas (such as roads, urban settlements, etc.). should deserve also attention in terms of modification of the processes of nutrient leaching/ retention. There is an abundant bibliography about these issues. Like I pointed out before, sampling took place along a significant period but nothing was observed about the temporal variation. Besides, there is also not a reference about the limitations of the methodology used (is it enough the establishment of boreholes only10 and 20m from the river?). Something that also needs attention is the relation between stream water quality and the concentration of nutrients in the boreholes.
Author Response
Dear Editor in Chief,
We highly appreciate your comments and advice, which gave us a chance to improve our manuscript. According to your comments, we have made the edits in the manuscript which are summarized in a table below. We also detected some issues which were improved in the revised version or the manuscript and are reported below, too.

Reviewer 2 Report (New Reviewer)
In this manuscript, the authors studied the impacts of organic and intensive agricultural production on 2 groundwater and surface water quality. Overall, the manuscript has a concrete structure, with promising results and discussions. However, there are few comments to be addressed.
Specific:
1. Abstract: Must be improved. Highlight the gap and also your novelty in the abstract.
2. Introduction section: The research gap and the research objectives were not clear in the submission. The application of modelling analysis has been performed extensively in this research direction; therefore, a proper literature review/table should be provided. Also, some relevant studies which can be included:
(i) Wong, Y.J., Shimizu, Y., He, K. et al. Comparison among different ASEAN water quality indices for the assessment of the spatial variation of surface water quality in the Selangor river basin, Malaysia. Environ Monit Assess 192, 644 (2020). https://doi.org/10.1007/s10661-020-08543
(ii) Wong, Y. J., Shimizu, Y., Kamiya, A., Maneechot, L., Bharambe, K. P., Fong, C. S., et al. (2021). Application of artificial intelligence methods for monsoonal river classification in Selangor river basin, Malaysia. Environmental Monitoring and Assessment, 193(7), 438, doi:10.1007/s10661-021-09202-y.
3. Methodology section: Provide a clear flowchart to ease the readers. Study area map is quite confusing at the moment. Suggest revising it. Better legend is necessary.
4. Explain the water quality classes. It will ease the readers on how this is going.
5. The comparative analysis is slightly confusing yixia.
6. Do include a seperate discussion section for more indepth discussion. You may compare it with other people work.
7. Conclusion section seems to be repetition of the results section. Huge modifications are required. Please make sure your ‘conclusion’ section underscores the scientific value added of your paper, and/or the applicability of your findings/results, as indicated previously. Please revise your conclusion part into more detail. Basically, you should enhance your contributions, limitations, underscore the scientific value added of your paper, and/or the applicability of your findings/results and future study in this section.
8. Implications for future research may also be included in the conclusion at the end.
Author Response
Dear Editor in Chief,
We highly appreciate your comments and advice, which gave us a chance to improve our manuscript. According to your comments, we have made the edits in the manuscript which are summarized in a table below. We also detected some issues which were improved in the revised version or the manuscript and are reported below, too.

Reviewer 3 Report (New Reviewer)
The article deals with the important issue of water quality, especially in areas with intensive agricultural activities. The article compiles an interesting set of data and uses analytical tools appropriate for this type of research. I recommend the article for publication after taking into account the following comments:
Abstract:
Some results are obvious without doing advanced research:
Line 22-25: „Assessing the nutrient dispersion from organic and intensive farming activities, almost all ground water samples collected from organic farm wells had lower nutrient content than the groundwater samples collected from intensive farming wells”
Materials and methods:
Unfortunately, nothing is known about the hydrological characteristics of the five rivers in question. What is their flow rate? What is the thermal regime? What effect does this have on water quality parameters?
How do the analyzed years compare to multi-year periods? Was the precipitation above normal or not? Were there droughts? What effect did this have on water flow and pollutant migration?
Figure 1. Please summarize the use structure of the analyzed polygons.
Results:
The Results chapter is too elaborate compared to the other chapters-at times monotonous in perception. Please consider a different form of presentation of some of the results (tables, figures).
Please explain the inaccuracy:
Line: 138-141: „Water research in 5 rivers located in the Nemunas River Basin District (Lomena and Romatas Kaišiadorys District, ŽemÄ—plėža KÄ—dainiai District, Šuoja Panevėžys District, and from Alšia River Prienai District) was conducted every month from 2014 to 2016”.
And then comes the sentence:
Line 184-201: „ The results for the Alšia River were good in 2012–2013….”
Discussion:
Please relate your own results to other such studies. This is currently lacking.
Author Response
Dear Editor in Chief,
We highly appreciate your comments and advice, which gave us a chance to improve our manuscript. According to your comments, we have made the edits in the manuscript which are summarized in a table below. We also detected some issues which were improved in the revised version or the manuscript and are reported below, too.

Round 2
Reviewer 1 Report (Previous Reviewer 3)
The discussion was considerably improved. However, it should be included a convenient justification of the establishment of the boreholes (terrain limitations, etc.)
Reviewer 2 Report (New Reviewer)
The authors have adopted the required changes substantially. The current manuscript is in good shape and is ready for publication. Therefore, I would recommend accepting the manuscript.
This manuscript is a resubmission of an earlier submission. The following is a list of the peer review reports and author responses from that submission.
Round 1
Reviewer 1 Report
I found this article interesting and indeed highlighting a knowledge gap of importance. However, there are many errors about formatting and typo. Moreover, results, discussion and conclusion part is not well written. Therefore, I do not recommend this article to be published in this Journal.
Reviewer 2 Report
Lines 119-121: Please change “P total” and “N total” to “total P” and “total N”.
Line 32: Only EU member states? Or over the globe? Generally, agriculture is a main driver of poor water quality across the world (see “Iran’s agriculture in the Anthropocene”).
Introduction: Can you please give more explanation about the human health consequences of nitrate such as blue baby, risk of cancers (see, “A non-threshold model to estimate carcinogenic risk of nitrate-nitrite in drinking water”)?
Lines 113-120: Please better justify the criteria used to select the study sites.
Figure 1: Please replace it with a high resolution figure. Also, the figure caption is not informative.
Figure 2: I couldn’t understand this figure. Can you please add more description to the manuscript?
Section 2.2: QA/QC is missing.
Figure 3: What do you mean by mg/IP and mg/IN in vertical axes?
Figure 4: Please change “mg/l” to “mg/L”.
Figure 5: see my previous comment.
Please rewrite conclusion.
Reviewer 3 Report
General comments: The idea of this paper is interesting, like the assessment of nutrient dispersion from organic and intensive farming activities, the design is correct since sampling points in the different rivers follow the same approach and the distribution of the dominant landscape types is relatively similar. The number of samples and the period of observation could also allow extracting conclusions about the impact on superficial and groundwater from organic or intensive agricultural areas. However, serious gaps make this manuscript scientifically relatively irrelevant. The conclusions are somewhat obvious, namely that P total, N total, NO3-N, NH4-N and PO4-P in the groundwater from organic farms were lower compared to the groundwater from intensive farming areas. What is after all the innovative character of this paper? The discussion is also virtually absent.
Here is a more detailed analysis:
Introduction: the pathways and origin of accumulation of N in groundwater are not described, the contrary of superficial water.
Results: It's more or less obvious the comparison between organic and intensive agriculture. More interesting would be the comparisons between nutrient concentrations 10 and 20 m from the river in both types of soil use.
Discussion: this is not a real discussion...it lacks an analysis of the chemical and physical processes that retain nutrients, and the disparity between organic and intensive farming relative to the increase or decrease of concentrations closer or farther away from the river. A comparison with other works is also essential. Data displayed a distinct spatial variation in nutrient concentration in organic and intensive farming but this didn't deserve attention from the authors. Moreover, nothing is mentioned about the techniques that can be used in intensive farming to reduce the eutrophication of superficial waters, this is the contribution of this paper to the appropriate management and rehabilitation of streams impacted by intensive farming. Aspects like the type of soils, riparian vegetation and slope/topography should deserve also attention. Sampling took place along a significant period but nothing was observed about the temporal variation. In fact, it was conducted a monitoring every month, for 3 years, with a collection of superficial and groundwater samples, but there is no observation of the temporal pattern.
Other aspects
Line 14 It’s not correct that the effects on groundwater by agriculture have not been studied: there is a large amount of literature about this subject.
Line 18. Ecological status is defined by biological, chemical and hydromorphological elements and not only by 2 or 3 chemical parameters, therefore it is wrong this classification according to WFD. This consideration appears everywhere in the text. The authors should just mention that the mentioned nutrients exceeded the water quality values…which is different from the ecological state. It is also not clear how it was established the physicochemical quality classes (ex. In Fig. 3).
Line 70 “increase increasing…” rephrase.
Line 220 I don't see this figure reported in the text…
Line 233 It's not correct that the boreholes are upstream or up to the river but 10 or 20 m far from the river since they are placed in a transversal section...this should be modified along the whole text.
Line 327. “The terrain had an effect.” Which effect?